# Preparation of Nanocellulose Using Ionic Liquids: 1-Propyl-3-Methylimidazolium Chloride and 1-Ethyl-3-Methylimidazolium Chloride

**DOI:** 10.3390/molecules25071544

**Published:** 2020-03-28

**Authors:** Marta Babicka, Magdalena Woźniak, Krzysztof Dwiecki, Sławomir Borysiak, Izabela Ratajczak

**Affiliations:** 1Department of Chemistry, Poznan University of Life Sciences, Wojska Polskiego 75, 60-625 Poznan, Poland; marta.babicka21@gmail.com (M.B.); magdalena.wozniak@up.poznan.pl (M.W.); 2Department of Food Biochemistry and Analysis, Poznan University of Life Sciences, Mazowiecka 48, 60-623 Poznan, Poland; krzysztof.dwiecki@up.poznan.pl; 3Institute of Chemical Technology and Engineering, Poznan University of Technology, Berdychowo 4, 60-965 Poznan, Poland; slawomir.borysiak@put.poznan.pl

**Keywords:** nanocellulose, ionic liquids, Fourier transform infrared spectroscopy, scanning electron microscopy, transmission electron microscopy, dynamic light scattering, X-ray diffraction

## Abstract

Cellulose nanocrystals were prepared using ionic liquids (ILs), 1-ethyl-3-methylimidazolium chloride [EMIM][Cl] and 1-propyl-3-methylimidazolium chloride [PMIM][Cl], from microcrystalline cellulose. The resultant samples were characterized by X-ray diffraction (XRD), Fourier transform infrared spectroscopy (FTIR), dynamic light scattering (DLS), scanning electron microscopy (SEM), and transmission electron microscopy (TEM). The XRD results showed that nanocellulose obtained by treatment with both ILs preserved basic cellulose I structure, but crystallinity index of samples (except for Sigmacell treated with [EMIM][Cl]) was lower in comparison to the starting microcrystalline cellulose. The DLS results indicated noticeably smaller particle sizes of prepared cellulose for material treated with [PMIM][Cl] compared to cellulose samples hydrolyzed with [EMIM][Cl], which were prone to agglomeration. The obtained nanocellulose had a rod-like structure that was confirmed by electron microscopy analyses. Moreover, the results described in this paper indicate that cation type of ILs influences particle size and morphology of cellulose after treatment with ionic liquids.

## 1. Introduction

Nanocellulose is a natural nanomaterial and has gained attention from researchers during the past decade due to its unique properties, such as light weight, high surface area to volume ratio, and excellent mechanical properties [1,2,3,4]. Moreover, nanocellulose possesses modifiable properties due to abundant hydroxyl groups and presents environmental benefits such as biodegradability and biocompatibility [5,6]. Because of these properties, nanocellulose can be used in numerous and various applications, such as biomedical applications (including drug delivery, tissue engineering scaffolds, wound dressings, cartilage repair, and separation of biomolecules and cells), reinforcement and fillers in composites, optical and electronic devices, or mobile field [6,7,8,9,10,11,12].

Several methods of nanocellulose production from different cellulose sources have been developed, including acid hydrolysis, mechanical process, enzymatic hydrolysis, and hydrolysis using ionic liquids [13,14,15,16]. Ionic liquids (ILs) are salts that consist of an organic cation and an organic or inorganic anion. The interest in ionic liquids application in numerous areas is caused by their unique and attractive properties, such as high polarity, a melting point below 100 °C, electrical conductivity, extremely low values of their saturated vapor pressure, non-flammability, non-volatility, and recyclability [17,18]. Ionic liquids are used to prepare nanocellulose from cellulosic material and also to dissolve cellulose [8,19,20,21,22,23]. Literature data showed that both the cation and the anion of ionic liquids affect cellulose dissolution [24,25,26]. The ability of ILs to dissolve cellulose is attributed to the effective breakage of inter- and intra-molecular hydrogen bonds in cellulose, wherein hydroxyl protons and oxygen of cellulose interact with anions and cations, respectively [25,27]. The ILs with chloride ions have been widely used in cellulose dissolution [27,28,29]. Chloride anion is non-hydrated and can break the hydrogen bonds of a cellulose network without derivatizing it [25,26]. The ionic liquids based on imidazolium cations are the most popular solvents among ILs in the process of nanocellulose preparation and cellulose decomposition [30,31]. Erdmenger et al. [26] investigated the ability to dissolve cellulose using 1-alkyl-3-imidazolium based ionic liquids with alkyl chains from ethyl to decyl. The authors observed the strong odd–even effect of the alkyl chains on the dissolving of cellulose in the IL for chain lengths up to hexyl [26]. The most commonly used ionic liquids for dissolving cellulose are 1-butyl-3-methylimidazole chloride ([BMIM][Cl]), 1-allyl-3-methylimidazole chloride ([AMIM][Cl]), and 1-ethyl-3-methylimidazole chloride ([EMIM][Cl]), which have proven to be particularly good solvents for cellulose, lignin, and other natural polymers [32,33,34,35].

However, there are few literature reports regarding the preparation of cellulose with nanometric dimensions using ionic compounds [32,33]. Tan et al. [8] obtained high crystallinity nanocellulose using 1-butyl-3-methylimidazolium hydrogen sulfate ([BMIM][HSO_4_]), both as the solvent and catalyst. Mao et al. [34] prepared cellulose nanowhiskers by single-stage hydrolysis of microcrystalline cellulose with an aqueous solution of 1-butyl-3-methylimidazolium hydrogen sulfate ([BMIM][HSO_4_]). Iskak et al. [29] produced nanocrystal cellulose from microcrystalline cellulose by hydrolysis in 1-butyl-3-methylimidazolium chloride ([BMIM][Cl]). The methods for preparing nanocellulose are constantly being improved, and new ionic liquids are used in this process. To the best of the authors’ knowledge, 1-propyl-3-methylimidazolium chloride [PMIM][Cl] has not been used in nanocellulose production. Moreover, there are also studies that describe the influence of different reaction conditions of nanocellulose preparation (including temperature, time of reaction, solvent, catalyst, or size of the cation of the imidazole ionic liquid) on properties of the obtained nanomaterial [24,29,35,36].

The aim of this work was to prepare nanocrystalline cellulose by treating microcrystalline cellulose with 1-ethyl-3-methylimidazolium chloride [EMIM][Cl] and 1-propyl-3-methylimidazolium chloride [PMIM][Cl]. The selection of ionic liquids was made on the basis of literature data, which indicate that cellulose dissolves effectively in ILs with cations having even numbered alkyl chains (such as 1-butyl-3-methylimidazolium chloride or 1-hexyl-3-methylimidazolium chloride) and is practically unable to dissolve in ILs with odd–even numbered alkyl chain substituent cations (such as 1-propyl-3-methylimidazolium chloride and 1-pentyl-3-methylimidazolium chloride) [19,26,37,38]. Moreover, literature data indicate that the length of an alkyl substituent in a cation affects the greenness rank of ionic liquids, and ILs with shorter alkyl chains are less toxic [39]. In addition, the type of cation may be decisive for the efficiency of cellulose hydrolysis, which is described in detail in [24]. Cation structure significantly affects distribution of particle sizes, which is key to obtaining a polymer filler with controlled properties. Therefore, in this study 1-ethyl-3-methylimidazolium chloride [EMIM][Cl] and 1-propyl-3-methylimidazolium chloride [PMIM][Cl] were used to determine their applicability in the production of nanocellulose.

Obtained samples of nanometric cellulose were characterized by Fourier transform infrared spectroscopy (FTIR), scanning electron microscopy (SEM), transmission electron microscopy (TEM), dynamic light scattering (DLS), and X-ray diffraction (XRD).

## 2. Results and Discussion

### 2.1. Characterization of Ionic Liquids

The first step of research was synthesis of the ionic liquids and a study of their characteristics using nuclear magnetic resonance (NMR) and FTIR. The results of NMR measurements for [EMIM][Cl] were as follows:

^1^H-NMR: δ (ppm) = 1.536 (t, 3H, CH_3_); 4.026 (s, 3H; CH_3_); 4.350 (q, 2H; CH_2_); 7.706 (t, 1H; CH); 7.751 (t, 1H; CH); 10.336 (s, 1H; CH);

^13^C-NMR: δ (ppm) = 14.135; 34.822; 43.532; 120.751; 122.269; 135.632.

The results of NMR measurements for [PMIM][Cl] were as follows:

^1^H-NMR: δ (ppm) = 0.940 (t, 3H; CH_3_); 1.938 (m, 2H, CH_2_); 4.050 (s, 3H; CH_3_); 4.268 (q, 2H; CH_2_); 7.734 (t, 1H; CH); 7.754 (t, 1H; CH); 10.346 (s, 1H; CH);

^13^C-NMR: δ (ppm) = 9.167; 22.101; 34.808; 49.674; 120.985; 122.276; 135.851.

Resonance signals from carbon coming from the imidazolium ring were found at approximately δ = 120, 122, and 135 ppm on the ^13^C-NMR spectra of both ionic liquids. The signal corresponding to the carbon in the first methyl group from the substituted alkyl chain was found at δ = 43.532 ppm in the spectrum of [EMIM][Cl] and δ = 49.674 ppm in the spectrum of [PMIM][Cl]. Subsequent carbon in the alkyl chain was registered at δ = 22.101 ppm and δ = 9.167 ppm for the propyl group present in [PMIM][Cl] and at δ = 14.135 ppm for the ethyl chain in [EMIM][Cl]. The highest values of resonance signals on the ^1^H-NMR spectra for both ionic liquids were found at approximately δ = 10.3 ppm and were assigned to the proton at the carbon between nitrogen atoms in the imidazolium ring. A triplet observed at δ = 7.75 ppm on the spectra of both ionic liquids corresponded to the remaining protons of the imidazolium ring. A singlet found at approximately δ = 4.0 ppm value on the spectra of both ionic liquids was assigned to the methyl group. Protons from the substituted alkyl chain from [PMIM][Cl] were registered at the values δ = 4.268 ppm as a quartet, δ = 1.938 ppm as a multiplet, and δ = 0.940 ppm as a triplet. In contrast, protons derived from the ethyl group of [EMIM][Cl] were observed at values δ = 4.350 ppm as a quartet and δ = 1.536 ppm as a triplet.

The FTIR spectra of synthesized ionic liquids and the spectrum of the synthesis substrate methylimidazole are shown in Figure 1.

In the spectra of both ionic liquids, there were observed bands at 2250, 1572, and 1175 cm^−1^, which were not registered in the spectrum of methylimidazole. The bands at the 2250 and 1572 cm^−1^ can be attributed to the stretching and deformation vibrations of positively charged nitrogen, respectively. The band in the region of 1170 cm^−1^ can be attributed to the C-C stretching vibration coming from the alkyl chain of ionic liquids. In the spectra, there are bands of ring stretching symmetries that occur at 1562 cm^−1^ (N=C) and 1450 cm^−1^ (CH_2_(N), CH_3_(N)CN). Moreover, there are observed peaks at 1085 cm^−1^, which could arise due to the N-C bond, and at 692 cm^−1^, which could be attributed to the C-H bond [40,41].

### 2.2. Characteristics of Cellulose Obtained by Hydrolysis in Ionic Liquids

#### 2.2.1. XRD Analysis

The next step of the research was to compare the methods used to characterize the nanocellulose. The XRD patterns of the cellulose samples before and after hydrolysis with ionic liquids are shown in Figure 2.

The diffraction pattern of all cellulose samples in Figure 2 showed only three peaks at 2θ = 15°, 17°, and 22.7° assigned to cellulose I. It was found that the diffraction pattern curves of the cellulose were characterized by variation in intensity of the (200) lattice plane of cellulose. During hydrolysis of cellulose in [EMIM][Cl], the intensity was higher compared to cellulose after hydrolysis with [PMIM][Cl]. This was due to the different course of the hydrolysis process, which consequently affects the supermolecular structure of cellulose. The decrease in the intensity of the maximum diffraction of the (200) plane indicates the destruction of crystallites caused by interaction with the ionic liquid. Much greater decrystallization occurred when using [PMIM][Cl], which proves that this ionic liquid hydrolysis was effective, caused degradation of crystal regions, and thus increased the content of amorphous phase. This is in line with the work of Li et al. [42], where the hydrolysis process is caused by the breaking of hydrogen bonds present in cellulose macromolecules, which as a consequence leads to shortening of polymer chains. Therefore, our observations regarding changes in the intensity of the diffraction maximum clearly indicate a different course of the cellulose hydrolysis process depending on the type of the cation. Further explanation is continued in the considerations regarding the determined values of the degree of crystallinity.

The calculated crystallinity index values of cellulose were 61% for untreated Avicel cellulose and 49% for untreated Sigmacell cellulose. The XRD results showed noticeable changes in crystallinity after the hydrolysis process. The crystallinity index of Avicel after reaction was 47% (after hydrolysis in [EMIM][Cl]) and 42% (after hydrolysis in [PMIM][Cl]). The results of the crystallinity index for Sigmacell were 53% (after hydrolysis in [EMIM][Cl]) and 36% (after hydrolysis in [PMIM][Cl]). It was found that the crystallinity index of both cellulose samples decreased when the 1-propyl-3-methylimidazolium chloride treatment was applied. The application of this ionic liquid to cellulose hydrolysis caused an approximate 30% decrease in the crystallinity index value. The obtained results were consistent with the work of Lee et al. [43], in which the authors described that the use of ionic liquids for cellulose hydrolysis in the initial phase of the process caused interaction with amorphous regions, thereby increasing the degree of crystallinity, but in subsequent stages, cellulose decomposed and the degree of crystallinity decreased. Mao et al. [35] observed that cellulose nanowhiskers obtained by single-stage hydrolysis of microcrystalline cellulose with an aqueous solution of 1-butyl-1-methylimidazolium hydrogen sulfate [BMIM][HSO_4_], exhibited diffraction peaks corresponding to crystallographic reflections of cellulose I, but the crystallinity degree of nanowhiskers was smaller compared to the starting material. The authors found that this phenomenon may result from swelling in the reaction medium and repeated centrifugation, a process known to reduce cellulose crystallinity index [34].

The hydrolysis process of cellulose in the ionic liquid with a shorter alkyl chain in the imidazolium ring ([EMIM][Cl]) had a different course in comparison to hydrolysis in [PMIM][Cl]. For Avicel cellulose, there was a decrease in the degree of crystallinity, but it was not as noticeable as for cellulose hydrolysis in [PMIM][Cl]. This may indicate a similar hydrolysis mechanism involving the interaction of an ionic liquid with amorphous and crystalline regions of cellulose. However, the interaction of [EMIM][Cl] with crystalline regions of Avicel cellulose may be more difficult and characterized by a less effective process of decrystallization than in the case of [PMIM][Cl]. In contrast, the crystallinity degree of Sigmacell cellulose treated with [EMIM][Cl] was higher than untreated material. This slight increase of crystallinity suggested that [EMIM][Cl] in the first step of hydrolysis eliminated small amounts of amorphous regions. The results of XRD analysis suggested that the introduction of a shorter alkyl chain into the ionic structure of the imidazolium liquid inhibited the interaction between the anhydro-glucopyranose rings and the cations of the ionic liquid. These results are in line with the results presented in work by Grząbka-Zasadzińska et al. [24], where the authors observed that a change of the imidazolium cation structure in ionic liquid is responsible for differences in the crystallinity degree of cellulose material after hydrolysis.

#### 2.2.2. FTIR Analysis

Figure 3 illustrates FTIR spectra of cellulose material before and after treatment with ionic liquids.

A broad band in the region of 3450 cm^−1^ observed in the spectra of all cellulose samples can be attributed to the characteristic hydrogen bond O–H stretching vibrations. Moreover, all samples showed a peak at 2900 cm^−1^ coming from sp^3^ C–H stretching vibration, and peaks at 1640 cm^−1^ corresponded to –OH bending of absorbed water [8,33,44]. The band observed at 1380 cm^−1^ in the spectra of Avicel and Sigmacell cellulose can be assigned to the C–O symmetric stretching and C–H bending bonds within the polysaccharide aromatic rings [8]. The band at 1060 cm^−1^ observed in the spectra of all cellulose samples can be attributed to the stretching vibration of the C–O–C pyranose ring of cellulose molecules [45]. In addition, the peaks found in the range of 1100 to 550 cm^−1^ were attributed to twisting, wagging, and deformation modes of anhydro-glucopyranose observed in the spectra of cellulose treated with [PMIM][Cl] and [EMIM][Cl]; these patterns are characteristic of β-glucosidic linkages [8,32,46]. Moreover, the changes in the spectra of cellulose (both in the form of Avicel and Sigmacell) treated with [PMIM][Cl] were more intensive than in the spectra of cellulose treated with [EMIM][Cl], which suggests that hydrolysis of cellulose with [PMIM][Cl] is more effective. The peak at 2900 cm^−1^ originating from sp^3^ C–H stretching vibrations is the weakest for the Sigmacell treated with the [PMIM][Cl] (F) sample, for which the lowest degree of crystallinity was also observed.

#### 2.2.3. DLS Analysis

The average particle size (hydrodynamic diameter) of the cellulosic material obtained as a result of the hydrolysis reaction with ionic liquids was evaluated using dynamic light scattering. The average particle size of microcrystalline cellulose (Avicel and Sigmacell) and cellulose after hydrolysis is presented in Figure 4.

Noticeably smaller particle sizes of prepared cellulose were detected for material treated with [PMIM][Cl] than cellulose hydrolyzed with [EMIM][Cl]. Cellulose hydrolyzed with [PMIM][Cl] was characterized by one narrow and well-defined peak, which indicated the size of particles was approximately 20 nm for Avicel and approximately 100 nm for Sigmacell. For Avicel microcrystalline cellulose treated with [EMIM][Cl], a well-distinguished peak in the particle size ranging from 480 to 1000 nm was observed. Additionally, a small peak assigned to the particle size of 70 to 100 nm was noted. For Sigmacell cellulose after hydrolysis with [EMIM][Cl], a single peak was observed in the range of 400 to 1000 nm. The particle size of Sigmacell cellulose after treatment with [EMIM][Cl] was slightly larger than the particle size of untreated cellulose (particles size in the range from 300 to 800 nm), which suggested that treated cellulose formed agglomerates, and thus one high and broad peak of ~1000 nm was observed. The possibility of agglomerates formation by cellulose treated with ionic liquids was also described by other authors [4,24]. The results of DLS analyses clearly indicate that cation type of ILs influences particle size of cellulose after treatment with ionic liquids. The obtained results were in agreement with the work of Grząbka-Zasadzińska et al. [24], who noticed influence of imidazole cation structure on the efficiency of the cellulose hydrolysis process and consequently on the particle size obtained. Moreover, based on the results of DLS (Figure 4) and XRD (Figure 2) analyses, it can be noted that cellulose (Avicel and Sigmacell) treated with [PMIM][Cl] had lower particle sizes and crystallinity index in comparison to [EMIM][Cl]-treated cellulose. These studies confirmed that for this type of ionic liquid, the process of cellulose hydrolysis took place both in amorphous and crystalline areas, which resulted in obtaining small values of crystallinity degree and consequently cellulose particles with a large amount of nanometric fraction. For ionic liquid containing an ethyl substituent in the imidazole cation structure, hydrolysis was not sufficient to obtain a noticeable nanometric fraction, which perfectly correlated with the diffractometric results for which no or an unnoticeable decrease in crystallinity was observed. Crystallinity degree of Sigmacell cellulose treated with 1-ethyl-3-methylimidazolium chloride that showed a tendency to agglomerate was slightly higher than for untreated cellulose.

#### 2.2.4. SEM and TEM Analyses

The morphology of cellulose treated with ILs was analyzed using SEM, and the results of this analysis in the form of images are presented in Figure 5. The SEM images indicated that hydrolysis of cellulose with both ionic liquids caused a reduction of raw cellulose fiber diameter. Moreover, cellulose particles after hydrolysis had a more irregular shape and more asymmetrical edges than untreated material. It is difficult to clearly discern individual whiskers from the agglomerated structure of obtained cellulose. The formation of agglomerates by IL-treated cellulose was especially well observed in cellulose hydrolyzed with [EMIM][Cl], which confirmed the results of DLS analysis. The SEM images of cellulose treated with 1-buthyl-3-methylimidazolium hydrogen sulphate ionic liquids presented in work by Man et al. [32] indicated that nanocrystal cellulose with a needle-like structure also formed agglomerates.

The results of TEM analysis (Figure 6) indicated that cellulose treated with both ionic liquids had a rod-like structure (except for Sigmacell treated with [EMIM][Cl]), in comparison to a more spherical structure of raw cellulose. The rod-like morphology of treated cellulose is similar to the structure of nanocellulose prepared using ionic liquids and presented in studies by Miao et al. [47] and Tan et al. [8]. Moreover, the TEM images also showed that IL-treated cellulose had a tendency to form aggregates, especially for [EMIM][Cl]-treated cellulose, which was in agreement with the results of DLS and SEM analyses.

## 3. Materials and Methods

### 3.1. Materials

Microcrystalline cellulose: Avicel PH-101 (~50 μm particle size) and Sigmacell Cellulose Type 20 (particle size equal 20 μm) were purchased from Sigma-Aldrich Chemie GmbH (Darmstadt, Germany). The substrates (1-methylimidazole, 1-chloroetane, and 1-chloropropane) and solvents (acetonitrile and heptane) for ionic liquids synthesis were of analytical grade and purchased from Sigma-Aldrich Chemie GmbH (Darmstadt, Germany). In all experiments, deionized water of Merck Millipore grade was used.

#### 3.1.1. Synthesis of the Ionic Liquids

Both ionic liquids were synthesized in a one-step reaction. The reaction substrates of the first ionic liquid synthesis were 1-methylimidazole, 1-chloroetane, and acetonitrile at a molar ratio of 1:1:10. The 1-methylimidazole, 1-chloropropane, and acetonitrile at a molar ratio of 1:1:10 were used to synthesize the second ionic liquid. Both mixtures were stirred under a reflux condenser at 90 °C for 24 h. The products of reactions were washed with heptane at 70 °C. Next, the solvent was removed using a rotary evaporator (BUCHI Labortechnik AG, Flawil, Switzerland). The final products of synthesis reactions were 1-ethyl-3-methylimidazolium chloride [EMIM][Cl] and 1-propyl-3-methylimidazolium chloride [PMIM][Cl]. The scheme of the ionic liquids synthesis is shown in Figure 7.

#### 3.1.2. Preparation of Cellulose Nanocrystals

Microcrystalline cellulose samples, Avicel and Sigmacell (0.2 g), were mixed with both synthesized ionic liquids, [EMIM][Cl] and [PMIM][Cl] (5 g), obtaining a 4% *w/w* ratio. The reactions were performed for 12 h at 100 °C, under intense stirring using a heating mantle with magnetic stirring (ChemLand, Stargard, Poland). The reactions were carried out without solvent and were quenched by adding 15 g of cold deionized water to the reaction mixtures to complete the reaction. The addition of water resulted in the formation of nanocellulose, which is insoluble in this solvent. The products of reactions were washed with acetonitrile. Nanocellulose was separated from the solution by centrifugation (6000 rpm for 30 min at room temperature) (Universal 320, Andreas Hettich GmbH & Co. KG, Tuttlingen, Germany) and after decantation, the nanomaterial was filtered through filter paper (Grade-5,Whatman) and dried initially at room temperature and finally over P_2_O_5_ (Sigma-Aldrich Chemie GmbH, Darmstadt, Germany). The methodology of nanocellulose preparation was based on the research described by Tan et al. [8] and Mao et al. [35].

### 3.2. Methods

#### 3.2.1. ^1^H and ^13^C Nuclear Magnetic Resonance Spectroscopy

The synthesized ionic liquids were characterized by nuclear magnetic resonance (NMR). The ^1^H-NMR and ^13^C-NMR experiments were performed on a VNMRS-400 spectrometer (Agilent Technologies, Santa Clara, CA, USA) operating at 402.644 MHz for ^1^H and 101.254 MHz for ^13^C. The tetramethylsilane (TMS) was used as a standard, and deuterated chloroform (CDCl_3_) was applied as a solvent for ionic liquids.

#### 3.2.2. FTIR Spectroscopy

Fourier transform infrared spectroscopy was used to characterize synthesized ionic liquids and determine the chemical structure of cellulose treated with these ionic liquids. Spectra were registered at a range of 4000 to 500 cm^−1^, at a resolution of 2 cm^−1^, and registering 16 scans using a Nicolet iS5 spectrophotometer (Thermo Fisher Scientific, Waltham, MA, USA). The cellulose samples were mixed with KBr (Sigma-Aldrich Chemie GmbH, Darmstadt, Germany) at a 1:200 mg ratio in the form of pellets and were analyzed by FTIR; the ionic liquids were determined in the form of liquid films.

#### 3.2.3. XRD Analysis

The supermolecular structure of cellulose samples treated with ionic liquids was analyzed by XRD. The diffraction pattern was recorded between 5° and 30° (2θ- angle range) in the step of 0.04°/3 s. The wavelength of the Cu K α radiation source was 1.5418 Å, and the spectra were obtained at 30 mA with an accelerating voltage of 40 kV. Deconvolution of peaks was performed by the method proposed by Hindeleh and Johnson [48] and improved and programmed by Rabiej [49]. After separation of XRD lines, the degree of crystallinity (X_c_) by comparison of areas under crystalline peaks and amorphous curve was determined.

#### 3.2.4. DLS Analysis

The particle size (hydrodynamic diameter) of cellulose treated with ionic liquids was determined using the DLS method. The cellulose samples were previously mixed with deionized water, treated by an ultrasound system (Sonic-2; Polsonic Palczyński Sp. J., Warsaw, Poland) for 25 min and then centrifuged using a Universal 320/320R centrifuge (Andreas Hettich GmbH & Co. KG, Tuttlingen, Germany) to remove the micrometric fraction of cellulose. The characterization of the hydrodynamic diameter of samples was performed using a Zetasizer Nano ZS-90 (Malvern Instruments Ltd., Malvern, UK) at room temperature, and the results are presented as size distribution by intensity.

#### 3.2.5. SEM Analysis

The morphology of samples was examined by a Zeiss EVO 40 scanning electron microscope (Carl Zeiss AG, Oberkochen, Germany). Before microscope analysis, cellulose samples were coated with a layer of gold using a Balzers SCD 00 sputter coater (BalTec Maschinenbau AG, Pfäffikon, Switzerland).

#### 3.2.6. TEM Analysis

The drops of diluted cellulose samples were applied on nickel mesh covered with a carbon film and allowed to dry. The TEM measurements were performed with a Hitachi HT7700 microscope (Hitachi High Technologies, Krefeld, Germany) operating at 100 kV.

## 4. Conclusions

Treatment of microcrystalline cellulose with ionic liquids 1-ethyl-3-methylimidazolium chloride [EMIM][Cl] and 1-propyl-3-methylimidazolium chloride [PMIM][Cl] resulted in obtaining nanocrystal cellulose. The XRD results showed that nanocellulose obtained by treatment with both ILs preserved basic cellulose I structure, but the crystallinity index of samples (except for Sigmacell treated with [EMIM][Cl]) was lower in comparison to the starting microcrystalline cellulose. The results of FTIR analysis also confirmed that the basic structure of nanocellulose was maintained. The DLS results indicated noticeably smaller particle sizes of prepared cellulose for material treated with [PMIM][Cl] compared to cellulose hydrolyzed with [EMIM][Cl]. The average particle size of Avicel cellulose treated with [PMIM][Cl] was approximately 20 nm, and for [PMIM][Cl]-treated Sigmacell cellulose it was approximately 100 nm. In turn, cellulose after hydrolysis in ionic liquid with a shorter alkyl chain formed agglomerates, which was also confirmed by SEM and TEM analyses. Moreover, the electron microscopy analyses indicated that the resultant samples had a rod-like structure. The results described in this paper indicate that cation type of ILs had an influence on particle size and morphology of cellulose after treatment with ionic liquids.

The presented results indicate that 1-propyl-3-methylimidazolium chloride [PMIM][Cl], which was used for the first time to obtain nanocellulose, is effective in this process. Moreover, the presented data demonstrate that in process of nanocellulose preparation, the ionic liquid with odd–even numbered alkyl chain substituent cations is more effective than IL with cation possessing even numbered alkyl chains. In addition, the results obtained are perfectly in line with previous work in this area. It has been shown that the structure of the ionic liquid, in particular the cation type, has a decisive influence on the course of cellulose hydrolysis and, consequently, on the obtained size of cellulose particles.

## Figures and Tables

**Figure 1 molecules-25-01544-f001:**
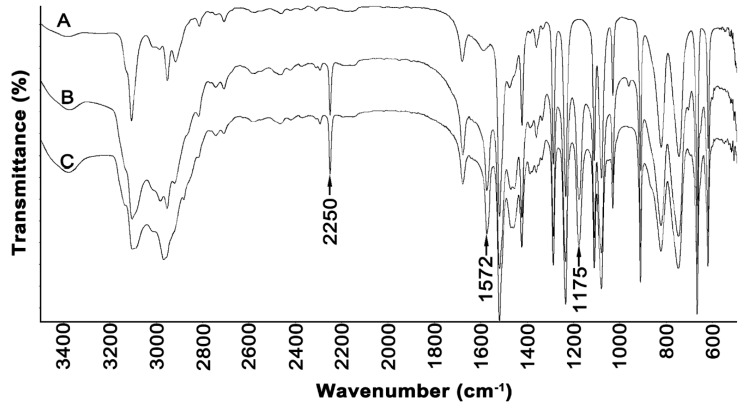
FTIR spectra of (**A**) methylimidazole, (**B**) [EMIM][Cl], and (**C**) [PMIM][Cl].

**Figure 2 molecules-25-01544-f002:**
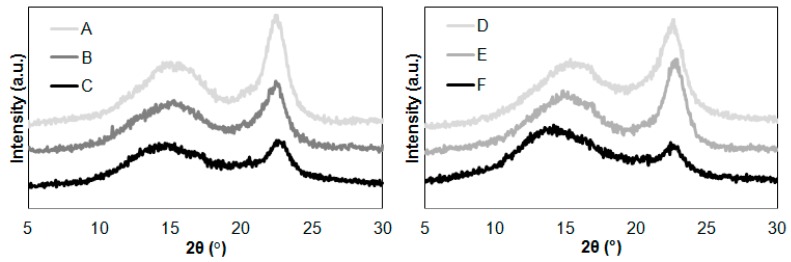
XRD patterns of cellulose after hydrolysis reaction with ionic liquids: (**A**) Avicel, (**B**) Avicel treated with [EMIM][Cl], (**C**) Avicel treated with [PMIM][Cl], (**D**) Sigmacell, (**E**) Sigmacell treated with [EMIM][Cl], and (**F**) Sigmacell treated with [PMIM][Cl].

**Figure 3 molecules-25-01544-f003:**
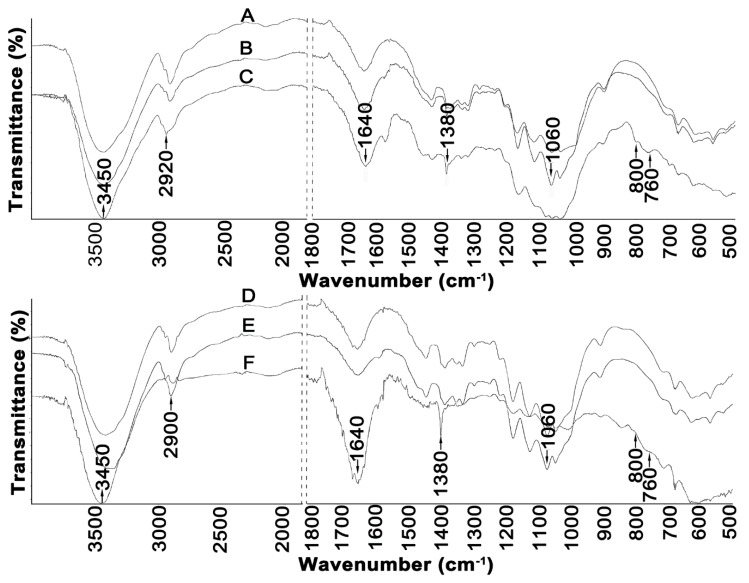
FTIR spectra of: (**A**) Avicel, (**B**) Avicel treated with [EMIM][Cl], (**C**) Avicel treated with [PMIM][Cl], (**D**) Sigmacell, (**E**) Sigmacell treated with [EMIM][Cl], and (**F**) Sigmacell treated with [PMIM][Cl].

**Figure 4 molecules-25-01544-f004:**
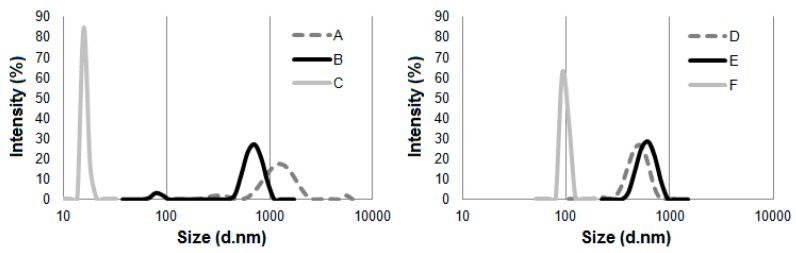
The average particle size of the cellulose before and after treatment with ionic liquids: (**A**) Avicel, (**B**) Avicel treated with [EMIM][Cl], (**C**) Avicel treated with [PMIM][Cl], (**D**) Sigmacell, (**E**) Sigmacell treated with [EMIM][Cl], and (**F**) Sigmacell treated with [PMIM][Cl].

**Figure 5 molecules-25-01544-f005:**
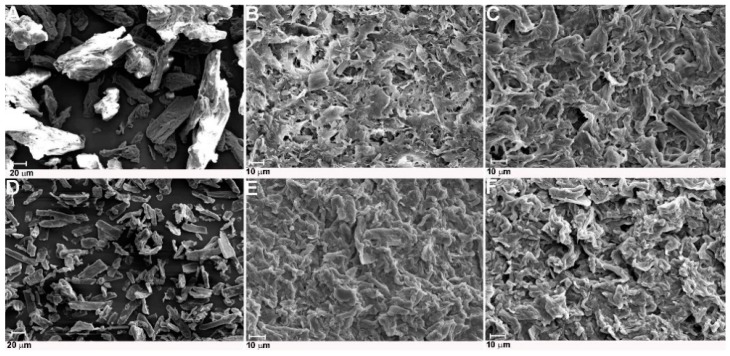
SEM images of (**A**) Avicel, (**B**) Avicel treated with [EMIM][Cl], (**C**) Avicel treated with [PMIM][Cl], (**D**) Sigmacell, (**E**) Sigmacell treated with [EMIM][Cl], and (**F**) Sigmacell treated with [PMIM][Cl].

**Figure 6 molecules-25-01544-f006:**
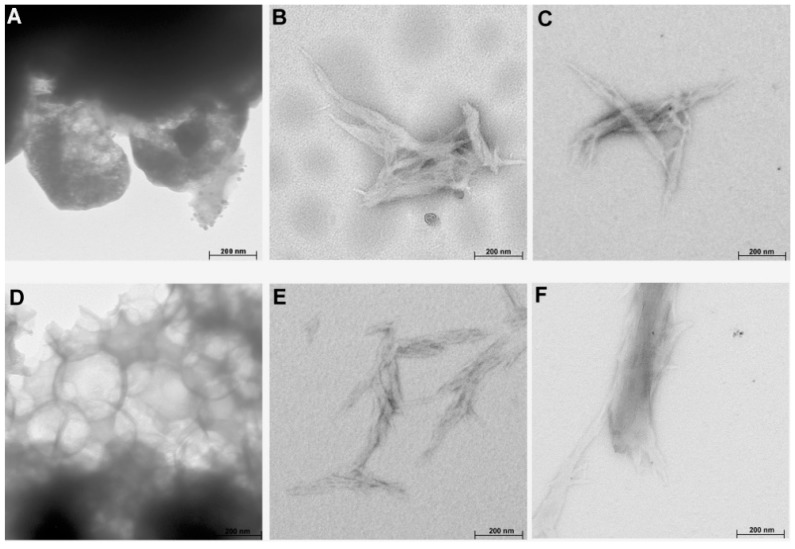
TEM images of (**A**) Avicel, (**B**) Avicel treated with [EMIM][Cl], (**C**) Avicel treated with [PMIM][Cl], (**D**) Sigmacell, (**E**) Sigmacell treated with [EMIM][Cl], and (**F**) Sigmacell treated with [PMIM][Cl].

**Figure 7 molecules-25-01544-f007:**
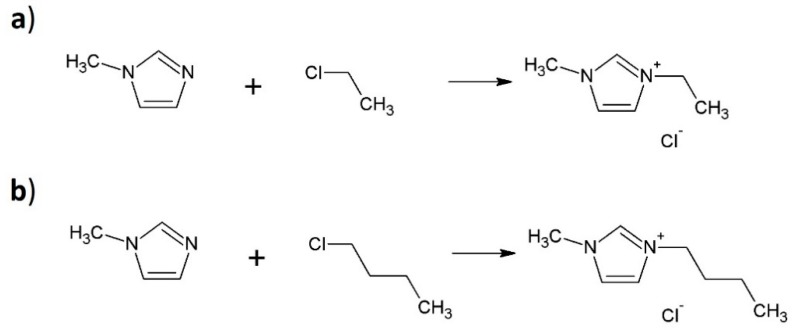
Synthesis of 1-ethyl-3-methylimidazolium chloride (**a**) [EMIM][Cl] and (**b**) 1-propyl-3-methylimidazolium chloride [PMIM][Cl].

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
