# Peer review of "Preparation of Nanocellulose Using Ionic Liquids: 1-Propyl-3-Methylimidazolium Chloride and 1-Ethyl-3-Methylimidazolium Chloride"

_molecules, 2020, doi:10.3390/molecules25071544_

Round 1

Reviewer 1 Report

Authors present producing of nanocrystalline cellulose from microcrystalline cellulose using treatment with two ionic liquids 1-ethyl-3-methylimidazolium chloride [EMIM][Cl] and 1-propyl-3- methylimidazolium chloride [PMIM][Cl]. The topic is in high interest and results have rather high scientific impact.

Authors synthesized ionic liquids on-site, that add value to the scientific work. NMR and FTIR results are presented in manuscript, however described results need to be supplemented with statement and proofs about purity of obtained ionic liquids in comparison with commercial ones. In FTIR spectra of ionic liquids there are lot of non-identified peaks (1600-600 cm-1), they should be explained.

Line 100. Figure 1. Change the axis titles. Transmittance should be given in % and Wavenumber in cm-1. The same for Figure 3. Change title axis!

Line 119. Statement “due to the course of the hydrolysis process in a different way” which was given as argument for different effect on supermolecular structure of cellulose, should be discussed in more details, when comparing treatment effect of ionic liquid on cellulose. What exactly was different in hydrolysis process?

Line 267. Authors say that nanocrystalline cellulose was filtered after obtaining by ionic liquid treatment. Could you please specify the filtration method, because it is not clear, what filtration methods can be used for nanoscale cellulose. Did all the initial material (microcrystalline cellulose) transformed into nanocrystalline cellulose and no separation of micro/nano phases was needed?

Author Response

Thank you for your helpful and constructive comments and suggestions. Below you will find our answers and replies to your points.

  1. Authors synthesized ionic liquids on-site, that add value to the scientific work. NMR and FTIR results are presented in manuscript, however described results need to be supplemented with statement and proofs about purity of obtained ionic liquids in comparison with commercial ones. In FTIR spectra of ionic liquids there are lot of non-identified peaks (1600-600 cm-1), they should be explained.

The NMR and FTIR spectra of 1-ethyl-3-methylimidazolium chloride [EMIM][Cl] were compared with the spectra available in database and literature (Lahiri A, Das R. (2012): Spectroscopic studies of the ionic liquids during the electrodeposition of Al-Ti alloy in 1-ethyl-3-methylimidazolium chloride melt, Materials Chemistry and Physics 132: 34-38). In turn, the NMR and FTIR spectra of 1-propyl-3-methylimidazolium chloride did not available in the spectral database, therefore the spectra of [PMIM][Cl] were compared with the spectra of 1-propyl-3-methylimidazolium bromide showed in literature (Kadari M., Belarbi E.H., Moumene T., Bresson S., Haddad B., Abbas Q., Khelifa B. (2017): Comparative study between 1-propyl-3-methylimidazolium bromide and trimethylene bis-methylimidazolium bromide ionic liquids by FTIR/ATR and FT RAMAN spectroscopies, Journal of Molecular Structure 1143: 91-99). Comparison of NMR and FTIR spectra of synthesized ionic liquids with available data indicates that obtained compounds were characterized by similar purity with compounds commercially available or obtained by other authors and presented in literature.

As you suggested, we supplemented the description of FTIR spectra in the article.

  1. Line 100. Figure 1. Change the axis titles. Transmittance should be given in % and Wavenumber in cm-1. The same for Figure 3. Change title axis!

Thank you for your observation. Our mistake have been corrected in the article.

  1. 3. Line 119. Statement “due to the course of the hydrolysis process in a different way” which was given as argument for different effect on supermolecular structure of cellulose, should be discussed in more details, when comparing treatment effect of ionic liquid on cellulose. What exactly was different in hydrolysis process?

We thank the Reviewer for these valuable comments. As suggested by the Reviewer, relevant changes have been made in the manuscript and are marked in yellow in the article.

  1. Line 267. Authors say that nanocrystalline cellulose was filtered after obtaining by ionic liquid treatment. Could you please specify the filtration method, because it is not clear, what filtration methods can be used for nanoscale cellulose. Did all the initial material (microcrystalline cellulose) transformed into nanocrystalline cellulose and no separation of micro/nano phases was needed?

Nanocellulose was separated from the solution by centrifugation (6,000 rpm for 30 minutes at room temperature) and after decantation, the nanomaterial was filtered through filter paper (Grade-5,Whatman) and dried initially at room temperature and finally over P2O5. This explanation has been added to the article. It is also worth emphasizing that such a filtration method is often used and described in many works e.g., Tan et al. [8] and Mao et al. [35].

Reviewer 2 Report

The manuscript by Ratajczak and coworkers reports the preparation of two ionic liquids with different imidazolium cations and their use for generation of nanocellulose. The nanocellulose samples are characterized using a number of standard techniques. The main finding is that the size and crystallinity are influenced by the cation (ie, ethyl vs methyl). Overall the findings are very much in line with several other recent (and more detailed) studies referenced in this paper. The use of the two ionic liquids does not appear to be particularly advantageous compared to the literature studies, given that the decreased crystallinity of the nanocelluloses. Specific comments are noted below.

  1. The last paragraph of section 1 should explain the motivation for examining other cations, beyond the fact that they have not yet been sued to generate nanocellulose.
  2. The discussion of the DLS data in Figure 4 should provide PdI as well as z-average.
  3. The TEM data in Figure 6 shows features for Avicel treated with [PMIM][Cl] that appear inconsistent with the small z-average shown in Figure 3.
  4. The conclusions fail to explain why the present observations are important, since they agree very well with previous work in this area.

Author Response

Thank you for your helpful and constructive comments and suggestions. Below you will find our answers and replies to your points.

1. The last paragraph of section 1 should explain the motivation for examining other cations, beyond the fact that they have not yet been sued to generate nanocellulose.

According to your suggestion, in last paragraph of section 1 we have added the explanation of ionic liquids selection.

  1. The discussion of the DLS data in Figure 4 should provide PdI as well as z-average.

The results of DLS measurements are related to the Brownian motion of the nanoparticles in the medium. The value obtained by this technique is the diameter (or radius) of a sphere having the same diffusion coefficient as particle investigated. The results of such measurements are most often presented in the form of particle size distribution plot. We have presented results in this way in our paper and it is presented as distribution plot in many other studies, such as:

  1. Arup Mandal, Debabrata Chakrabarty, Isolation of nanocellulose from waste sugarcane bagasse (SCB) and its characterization, Carbohydrate Polymers, Volume 86, Issue 3,2011, 1291-1299, https://doi.org/10.1016/j.carbpol.2011.06.030.
  2. Marcos Aurélio Dahlem, Cleide Borsoi, Betina Hansen, André Luís Catto, Evaluation of different methods for extraction of nanocellulose from yerba mate residues, Carbohydrate Polymers, Volume 218, 2019, 78-86, https://doi.org/10.1016/j.carbpol.2019.04.064.
  3. Fuge Niu, Mengya Li, Qi Huang, Xiuzhen Zhang, Weichun Pan, Jiansheng Yang, Jianrong Li, The characteristic and dispersion stability of nanocellulose produced by mixed acid hydrolysis and ultrasonic assistance, Carbohydrate Polymers, Volume 165, 2017,197-204, https://doi.org/10.1016/j.carbpol.2017.02.048.
  4. Matjaž Kunaver, Alojz Anžlovar, Ema Žagar, The fast and effective isolation of nanocellulose from selected cellulosic feedstocks, Carbohydrate Polymers, Volume 148, 2016, 251-258, https://doi.org/10.1016/j.carbpol.2016.04.076.
  5. Grząbka-Zasadzińska A., Klapiszewski Ł., Borysiak S., Jesionowski T., Functional MgO-lignin hybrids and their application as fillers for polypropylene composites, Molecules, 2020, 25, 864
  6. Grząbka-Zasadzińska, A.; Skrzypczak, A.; Borysiak, S. The influence of the cation type of ionic liquid on the production of nanocrystalline cellulose and mechanical properties of chitosan-based biocomposites. Cellulose 2019, 26(8), 4827-4840.
  7. Klapiszewski Ł., Grząbka-Zasadzińska A., Borysiak S., Jesionowski T., Preparation and characterization of polypropylene composites reinforced by functional ZnO/lignin hybrid materials, Polymer Testing, 2019, 79, 106058.
  8. Grząbka-Zasadzińska A., Klapiszewski Ł., Borysiak S., Jesionowski T., Thermal and Mechanical Properties of Silica-Lignin/Polylactide Composites Subjected to Biodegradation, Materials, 2018, 11, 2257.

Particle size distribution plot characterizes the sample in a way closer to reality than the z-average, because nanoparticles always have a diameter within a certain range. By analyzing the width and number of peaks on the particle size distribution plot, we can also obtain information on the sample polydispersity.

  1. The TEM data in Figure 6 shows features for Avicel treated with [PMIM][Cl] that appear inconsistent with the small z-average shown in Figure 3.

We are much grateful to the Reviewer for this comment and we appreciate his/ her knowledge allowing a broader approach to the problems studied. After analyzing the literature, we can assume that this “inconsistent” is the result of the methodology of the research used.

Dynamic Light Scattering is widely used to obtain the statistical size distribution of cellulose nanoparticles in suspension. It is worth emphasizing that size values depend on the orientation of fibers in suspension (Frone et al., 2011). Sizes reported are approximations of real size values, but are adequate for comparison studies among different samples (Frone et al., 2011).

  1. Frone, A. N., Panaitescu, D. N., Donescu, D., Spataru, C. I., Radovici, C., Trusca, R., et al. (2011). Preparation and characterization of PVA composites with cellulose nanofibrils obtained by ultrasonication. Bioresources, 6, 487-512.

Moreover, for DLS analysis, cellulose solutions were treated with ultrasound system. In turn, in TEM analysis, cellulose samples did not treated with ultrasound system and centrifuged because small amounts of cellulose samples used in this measurement. The cellulose samples were applied on nickel covered with a carbon film. Probably, such differences in methodology, could be responsible for the formation of agglomerates of nanometric cellulose particles, as observed in the Figure 6. Obtaining good quality of TEM images is difficult, especially in the case of cellulosic suspensions which has been described in several papers, e.g. Samir et.al., 2005.

  1. Samir et.al., Review of recent research into cellulosic whiskers, their properties and their application in nanocomposites field, Biomacromolecules, 2005, 6, 612-26.

Unfortunately, we do not have our own TEM microscope and we are not able to improve these images. Therefore, if the Reviewer decides that TEM images are unnecessary, we will of course remove them.

Moreover, the measurement methodology has been supplemented in the article.

  1. The conclusions fail to explain why the present observations are important, since they agree very well with previous work in this area.

As you suggested, additional sentences introduced into the revised version of our paper.

Reviewer 3 Report

The paper entitled “Preparation of Nanocellulose Using Ionic Liquids: 1-Propyl-3-Methylimidazolium Chloride and 1-Ethyl-3-Methylimidazolium Chloride” is an interesting manuscript I suggest the publication in Molecules.

In  this  research activity, the authors proposed the extraction and characterization of cellulose nanocrystals  using  ionic  liquids  (ILs),  1-ethyl-3-methylimidazolium chloride [EMIM][Cl] and 1-propyl-3-methylimidazolium chloride [PMIM][Cl], from  microcrystalline  cellulose.  The  materials  were  characterized  by  X-ray  diffraction (XRD), Fourier transform infrared spectroscopy (FTIR), dynamic light scattering (DLS), scanning electron microscopy (SEM), and transmission electron microscopy (TEM). The XRD results showed that nanocellulose obtained by treatment with both ILs preserved basic cellulose I structure, but  crystallinity  index  of  samples  (except  for  Sigmacell  treated  with  [EMIM][Cl])  was  lower  in  comparison to the starting microcrystalline cellulose. The DLS results indicated noticeably smaller  particle  sizes  of  prepared  cellulose  for  material  treated  with  [PMIM][Cl]  compared  to  cellulose  samples  hydrolyzed  with  [EMIM][Cl],  which  were  prone  to  agglomeration. The SEM and TEM investigations showed that nanocellulose  had  a  rod-like  structure. The authors observed that cation type of ILs influences particle size  and morphology of cellulose after treatment with ionic liquids.  

Specific comments:

Introduction-paragraph 1: The authors are invited to explain better the novelty proposed in this research activity.

Preparation of cellulose nanocrystals- paragraph 3.1.2: The authors are invited to rewrite this paragraph. Specifically, they are invited explain the use of 15 ml of cold water to stop the process  to which initial quantity of solution it refers. The authors are invited to write this quantity in grams.

Author Response

Thank you for your helpful and constructive comments and suggestions. Below you will find our answers and replies to your points.

1.Introduction-paragraph 1: The authors are invited to explain better the novelty proposed in this research activity.

As you suggested, the introduction in the article has been completed.

  1. Preparation of cellulose nanocrystals- paragraph 3.1.2: The authors are invited to rewrite this paragraph. Specifically, they are invited explain the use of 15 ml of cold water to stop the process  to which initial quantity of solution it refers. The authors are invited to write this quantity in grams.

As you suggested, the paragraph 3.1.2 was modified and the explanation of use of cold water has been added to this paragraph.

Round 2

Reviewer 2 Report

The authors have dealt adequately with my previous comments.  In particular the additions to the Introduction and Conclusions are useful in clarifying the goals of the study and how the conclusions complement the information already available in the literature.